# Preparation of Nitrogen and Sulfur Co-Doped Fluorescent Carbon Dots from Cellulose Nanocrystals as a Sensor for the Detection of Rutin

**DOI:** 10.3390/molecules27228021

**Published:** 2022-11-18

**Authors:** Tao Zhang, Qingxue Ji, Jiayi Song, Haiming Li, Xing Wang, Haiqiang Shi, Meihong Niu, Tingting Chu, Fengshan Zhang, Yanzhu Guo

**Affiliations:** 1Liaoning Key Laboratory of Lignocellulose Chemistry and BioMaterials, Liaoning Collaborative Innovation Center for Lignocellulosic Biorefinery, College of Light Industry and Chemical Engineering, Dalian Polytechnic University, Ganjingzi District, Dalian 116034, China; 2Key Laboratory of Pulp and Paper Science & Technology of Ministry of Education, Qilu University of Technology (Shandong Academy of Sciences), Changqing District, Jinan 250353, China; 3Huatai Group Corp. Ltd., Dongying 257335, China

**Keywords:** cellulose nanocrystals, carbon dots, hydrothermal method, fluorescent sensing

## Abstract

The poor water solubility, large particle size, and low accessibility of cellulose, the most abundant bioresource, have restricted its generalization to carbon dots (CDs). Herein, nitrogen and sulfur co-doped fluorescent carbon dots (N, S-CDs) were hydrothermally synthesized using cellulose nanocrystals (CNC) as a carbon precursor, exhibiting a small particle size and excellent aqueous dispersion. Thiourea was selected as a nitrogen and sulfur dopant to introduce abundant fluorescent functional groups into N, S-CDs. The resulting N, S-CDs exhibited nanoscale size (6.2 nm), abundant functional groups, bright blue fluorescence, high quantum yield (QY = 27.4%), and high overall yield (16.2%). The excellent optical properties of N, S-CDs endowed it to potentially display a highly sensitive fluorescence “turn off” response to rutin. The fluorescence response for rutin allowed a wide linear range of 0–40 mg·L^−1^, with a limit of detection (LOD) of 0.02 μM, which revealed the potential of N, S-CDs as a rapid and simple sensing platform for rutin detection. In addition, the sustainable and large-scale production of the N, S-CDs in this study paves the way for the successful high-value utilization of cellulose.

## 1. Introduction

Currently, carbon dots (CDs), as a new type of fluorescent carbon nanomaterial, are attracting increasing attention due to their favorable characteristics, e.g., excellent optical properties, low toxicity, good water solubility, and great biocompatibility [1,2]. Extensive studies have been focused on how to utilize CDs in the fields of biomedical imaging [3], fluorescence labeling [4], probes [5], and optical elements [6]. Up until to now, a variety of raw materials, including benzyl disulfide [7], citric acid [8], polyethyleneimine [9], and *o*-phenylenediamine [10], have been used as precursors to prepare CDs. Compared to the above listed raw materials, biomass materials are more favored due to their unique advantages, such as low cost, high yield, and high carbon and functional group contents [11,12].

Cellulose, as one of the most abundant biomass polymers on the earth, has been further employed as a carbon precursor to prepare CDs. Various sources of cellulose, including tree pulp [13], waste papers [14], and cellulose powders [15], have been utilized to prepare CDs. The method of hydrothermal treatment is commonly adopted due to its low cost and simplicity. Generally, the cellulose-based CDs emit blue fluorescence, with an emission wavelength between 380–460 nm and a QY value lower than 10% [16]. However, the natural properties of cellulose, e.g., high molecular weights, large particle size, and low accessibility [17,18], have limited the conversion of cellulose into CDs during hydrothermal treatment, which then generates CDs with low quantum yield (QY) and poor fluorescence properties [19]. In addition, the low yield and high residue rate of the CDs from cellulose limit the full utilization of cellulose. Therefore, researchers have focused their attention on using the water-soluble or water-dispersible cellulose derivatives as precursors. Carboxymethyl cellulose (CMC), cellulose nanocrystals (CNC), and hydroxypropyl cellulose (HPC) have been widely reported as precursors to prepare CDs [20,21]. CNC, a type of nanometer-sized biomass-derived material, has received increasing attention from both academic and industrial researchers due to its excellent properties, e.g., nanoscale diameter, excellent water solubility, high mechanical strength, and specific surface area [22,23]. Owing to its sufficient amounts of surface hydroxyl groups, the CNC can be easily functionalized and reacted with the passivators during hydrothermal treatment, which makes it to be an ideal material for the preparation of CDs.

In this work, CNC with a small particle size and excellent dispersion in aqueous solution was obtained by hydrolyzing cellulose with sulfuric acid. The as-prepared CNC was selected as a precursor, and thiourea was chosen as the dopant to prepare N, S-CDs via a facile hydrothermal treatment. The advanced carbon source and the doping of heteroatoms empowered the prepared N, S-CDs outstanding optical properties (QY = 27.4%). Notably, the yield of N, S-CDs reached 16.2% due to its high accessibility, small particle size, and good dispersion of CNC. Moreover, the N, S-CDs exhibited potential as a nanoprobe for the detection of rutin. Rutin, a widely used type of anticancer flavonoid, has attracted extensive attention in clinical applications [24]. However, an overdose of rutin will result in irreversible side effects, such as acute hepatic and renal damage [25]. Hence, the detection of rutin is of great significance. High-performance liquid chromatography [26], electrochemical detection [27], and capillary electrophoresis [28] are widely employed to detect rutin. The above methods require professional training, expensive instruments, and sophisticated procedures. In contrast to the above methods, fluorescence assays, with rapid response and simple operation, have drawn extensive attention. Nevertheless, the toxic and expensive fluorescence probes (e.g., dyes [29] and semiconductor quantum dots [30]) have limited the promotion of fluorescence assays. Hence, the construction of fluorescence sensing nanoprobes from environmentally friendly and inexpensive N, S-CDs would confirm its practical application value. In conclusion, this work offers a green, effective, and low-cost strategy to produce CDs, providing a novel insight into the high-value utilization of biomass.

## 2. Results and Discussion

### 2.1. Synthesis and Characterization of N, S-CDs

CNC is an advantageous carbon precursor due to its outstanding properties, such as large specific surface area, reactive surfaces, and nanoscale particle size. In this paper, cellulose was hydrolyzed with sulfuric acid to obtain CNC with a small particle size and good dispersion in aqueous solutions, according to previous report [31]. As shown in Appendix A, SEM and TEM were employed to characterize the micromorphology of CNC. The as-prepared CNC exhibited a needle-like shape with a diameter of 5–20 nm in width and 100–200 nm in length, consistent with previous reports [32]. Subsequently, CNC was chosen as a carbon source and subjected to a one-step hydrothermal process to prepare the N, S-CDs. During this process, thiourea was selected as a passivating agent to generate the N, S-CDs (Figure 1A). The QY value of N, S-CDs was selected as a parameter to evaluate the effect of preparation conditions (e.g., the temperature, reaction time, and mass ratio of thiourea/CNC) on the property of N, S-CDs. As shown in Figure 1B, the QY value of N, S-CDs gradually increased as the reaction temperature increased from 200 to 240 °C. However, further heating the temperature to 240 °C would cause a decline in QY value. In addition, the effect of reaction time on the QY value displays a similar change and tends to decrease the reaction temperature (Figure 1C). This phenomenon could be ascribed to the decrease in the surface fluorescent functional groups of N, S-CDs, which was caused by the increase in reaction temperature and time. As shown in Figure 1D, the optimized mass ratio of thiourea to CNC was 0.8:1. Based on above analyses, the highest QY value (27.4%) of N, S-CDs was observed under the conditions of 0.8:1 mass ratio of thiourea to CNC at 240 °C for 24 h.

As shown in Figure 2A, the N, S-CDs were well dispersed in an aqueous solution, with nearly spherical shape. The particle size of N, S-CDs ranged between 3 and 7.5 nm, and its average particle diameter was calculated to be 6.22 nm. The structures and surface groups of C-CDs and N, S-CDs were further characterized by FT-IR, XRD, and Raman techniques. Compared to the FT-IR spectrum of C-CDs in Figure 2B, the FT-IR spectrum of N, S-CDs exhibited the characteristic peaks of C-N (1410 cm^−1^), =NH^+^ (1556 cm^−1^), -SO_3_^−^ (1095 cm^−1^), and NH (804 cm^−1^) [33,34], confirming that the nitrogen and sulfur elements were successfully introduced into the structures of N, S-CDs. The XRD patterns (Figure 2C) of C-CDs and N, S-CDs both displayed a wide diffraction peak at 24.8°, which was similar to the XRD pattern of graphite crystals (002) [35]. The interplanar crystal spacing of the C-CDs was calculated to be 0.35 nm, while that of the N, S-CDs was 0.36 nm, which was due to the fact that the surface structure of the N, S-CDs was rich in O-H, C-N, and C=O functional groups [36,37]. However, the interplanar crystal spacings of the C-CDs and N, S-CDs were larger than those of the graphite crystal structure (0.34 nm) at the (002) peak. The Raman spectra of N, S-CDs are shown in Figure 3D. The D peak at 1352 cm^−1^ corresponds to the vibrations of the disordered sp^3^ hybridized carbon atoms. The G peak at 1568 cm^−1^ is associated with the vibrations of the ordered hybridized carbon atoms of sp^2^ [38,39]. The I_D_/I_G_ ratio (calculated by the area of the D peak to the G peak) was 0.99, which further indicated that the N, S-CDs exhibited a graphite structure [40].

In comparison with the XPS survey spectra of CNC and C-CDs (Figure 3A), the XPS spectrum of N, S-CDs displayed three new peaks at 396 eV (N1s), 150 eV (S2p), and 223 eV (S2s) [41,42], confirming the successful passivation of nitrogen and sulfur elements into N, S-CDs. The chemical compositions of CNC, C-CDs and N, S-CDs are shown in Table 1. The C/O value of CNC was 1.25, and the C/O values of C-CDs and N, S-CDs were enhanced to 2.29 and 2.17, respectively. The nitrogen and sulfur content in N, S-CDs was 7.90% and 6.96%, respectively. In the N1s high-resolution spectrum of N, S-CDs (Figure 3B), the characteristic peaks at 398.7, 399.8, and 400.6 eV are related to the -N=, C-N, and N-H groups, respectively [43]. In the S2p high-resolution spectrum of N, S-CDs (Figure 3C), there were six peaks locating at 161.5, 163.2, 164.6, 167.9, 168.7, and 169.3 eV, which are assigned to C-S, the C-S-C of S2p_3/2_, the C-S-C of S2p_1/2_, -C-S(O)_2_-C, -C-S(O)_3_-C-, and -C-S(O)_4_-C-, respectively. The high resolution C1s of N, S-CDs (Figure 3D) confirmed the existence of carbons in the forms of C-H/C-C, C-N/C-S, C-O, and C=O at 288.6, 284.5, 285.3, and 286.5 eV, respectively. As shown in Figure 3E, the forms of oxygen in N, S-CDs were C-O/C=O at 531.9 eV and C-OH/C-O-C at 538.8 eV [44].

Based on the above analyses, the chemical structure of N, S-CDs is illustrated in Figure 3F. The N, S-CDs possessed abundant surface functional groups (e.g., -C=O, -OH, -NH_2_, and -SO_x_), and a stable nanosized carbon core. In addition, the pyridine N and the graphite N were regularly doped in the skeleton of carbon core, which was caused by the doping of thiourea. The improved carbon skeleton and the changed surface state suppressed the π–π interactions between carbon cores, depressed nonradiative recombination electron-donating, and facilitated high-yield radiative recombination, which promoted the QY value of N, S-CDs [45].

### 2.2. Optical Properties and Fluorescence Stability of N, S-CDs

The UV-vis absorption and fluorescence spectroscopy were applied to analyze the optical properties of the N, S-CDs aqueous solution. As shown in the inset image in Figure 4A, the aqueous solution of N, S-CDs appeared yellowish in the daylight and emitted blue fluorescence under excitation of UV light at 365 nm (Figure 4A). The N, S-CDs solution showed two characteristic peaks at around 273 nm and 325 nm. The UV-vis absorption peak at 273 nm is assigned to the π–π* transition of the sp^2^ domains from the carbon core [46]. The absorption peak at 325 nm is corresponding to the n-π* transitions associated with the C=O bond [4,47]. Furthermore, the maximum excitation and emission wavelengths of the N, S-CDs solution were located at 325 and 405 nm. As demonstrated in Figure 4B, the fluorescence emission of N, S-CDs showed an excitation-dependent feature, which gradually red-shifted as the excitation wavelength was altered from 305 nm to 425 nm. This excitation-dependent fluorescence feature existed widely in the fluorescent carbon quantum dots, which might be due to the different particle sizes and functional surface groups [48].

In order to realize the maximum sensitivity of N, S-CDs, the effect of the pH value on the fluorescence intensity of N, S-CDs was analyzed. As shown in Figure 4C, the fluorescence intensity of N, S-CDs was stable in acidic and neutral pH environments, which reached a maximal value at pH 5.0. This phenomenon was related to the deepening of the deprotonation of the surface groups (-NH_2_ and -SO_x_) with the increase in pH value. The deprotonated surface functional groups further caused the changes in the fluorescence behavior of N, S-CDs [49]. Based on the above analysis, the solution with a pH value of 5.0 was selected for subsequent experiments. Figure 4D depicts the influence of sodium chloride ionic concentrations on the fluorescence intensity of N, S-CDs. It was observed that negligible changes in the fluorescence emission intensity of N, S-CDs were found by increasing the sodium chloride ionic concentrations from 0 mol/L to 1 mol/L. In addition, the photostability of N, S-CDs is shown in Appendix A. As the UV irradiation time increased from 0 to 180 min, the fluorescence intensity of N, S-CDs did not change significantly. The above results indicated that the N, S-CDs possessed excellent salt resistance and photostability.

### 2.3. Fluorescence “Turn off” Detection of Rutin

To achieve the maximum sensitivity of N, S-CDs to rutin, the equilibrium time after the addition of rutin (45 mg·L^−1^) was optimized. As shown in Appendix A, after the addition of rutin into the N, S-CDs solution, the fluorescence intensity of the N, S-CDs/rutin system was rapidly decreased and gradually became stable after 5 min. Therefore, 5 min was selected as the equilibrium time for the subsequent detection experiment. Typically, the presence of some additives in rutin tablets might have an impact on the rutin detection; therefore, we performed the investigation on the fluorescence intensity after adding different amino acids and metal ions simultaneously with the rutin. As shown in Figure 5A,B, the fluorescence properties of the N, S-CDs solution are quenched by rutin. In addition, the addition of other interfering substances, such as amino acids and metal ions, did not lead to an obvious fluorescence quenching response, indicating that N, S-CDs were highly selective for the fluorescence sensing system of rutin.

Figure 6A illustrates the fluorescence quenching process of N, S-CDs in the presence of rutin. As shown in Appendix A, the absorption peaks of rutin were obviously increased with the presence of N, S-CDs. The phenomenon could be ascribed to the interaction between rutin and N, S-CDs. The above characterization results, combined with previous reports, suggested that the mechanism of fluorescence quenching of N, S-CDs could be attributed to static quenching. The zeta potential of the fluorescence quenching system is depicted in Figure 6B. The zeta potential of N, S-CDs, rutin, and N, S-CDs in the presence of rutin were 1.27, −15.58, and −7.81 mV, respectively. The positive value of N, S-CDs could be attributed to the nitrogen-containing groups, i.e., amino groups, on the surface of N, S-CDs. The electric surface interactions between N, S-CDs and rutin were further confirmed by their opposite potential, which was consistent with the UV-vis characterization. The above analysis confirmed that the fluorescence quenching mechanism could be attributed to static quenching [50]. As shown in Figure 6C, the average lifetime of N, S-CDs in the absence and presence of rutin was 5.27 and 5.14 ns, respectively. These results indicated insignificant changes in the lifetime of N, S-CDs after the addition of rutin, which further confirmed the static quenching-dominated mechanism between the system. In order to verify that the quenching was not caused by dynamic quenching, we have measured the Stern–Volmer plots under different temperatures, and the results are shown in Appendix A. Typically, in the process of dynamic quenching, the fluorescence will be enhanced via increasing dynamic collision probability at high temperatures. As shown in Appendix A, the inverse correlation between quenching efficiency and temperature suggested that the probable quenching mechanism was not initiated by dynamic quenching [51]. It was observed from Figure 6D that the N, S-CDs solution emitted a strong blue fluorescence under 365 nm UV lamp irradiation, and the blue fluorescence was almost completely extinguished after the addition of rutin with a certain concentration. By increasing the concentration of rutin from 0 to 125 mg·L^−1^, the fluorescence intensity of N, S-CDs solution was gradually reduced, and the fluorescence emission peak was red-shifted. Figure 6E shows the relationship between F_0_/F value and the rutin concentration in the range of 0–125 mg·L^−1^. It fitted the linear Stern–Volmer equation with a correlation coefficient R^2^ = 0.9987 when the rutin concentration was in the range of 0 to 40 mg·L^−1^. The Stern–Volmer quenching constant was found to be 55,260 M^−1^, revealing the high quenching efficiency of N, S-CDs to rutin. In addition, the linear correlation between F_0_/F value and the rutin concentration confirmed that the interaction between rutin and N, S-CDs could be attributed to the static quenching, which was consistent with the above analysis [52]. In Table 2, the N, S-CDs prepared from the most abundant biomass material—cellulose—exhibited rutin detection performance comparable to the reported for optical nanomaterials. The LOD (0.02 μM) and K_SV_ of N, S-CDs to rutin were close to or better than the other reported metal-doped optical nanomaterials. The results showed that the N, S-CDs exhibited high sensitivity and selectivity to rutin, which was due to the selective interaction between the rutin and the surface functional groups of the N, S-CDs. The oxygen containing functional groups of the N, S-CDs could be reduced by the reducible rutin, which led to the fluorescence quenching of the N, S-CDs. Moreover, our facile approach for preparing N, S-CDs with excellent detection performance to rutin from green and economic cellulose materials would pave a way for the high value utilization of cellulose.

### 2.4. Practical Application in Real Samples

The N, S-CDs was employed to detect rutin in simulation environments to confirm its potential for practical application. The rutin content is calculated by the fluorescence intensity of real samples, and the formula is given in Figure 6C. The results of rutin recoveries in lake water, human urine, and serum samples are shown in Table 3. The measured results were close to those for the spiked rutin; the spiked recoveries were in the range of 96.3% to 104.3% and the RSD was below 3.2%. The real sample analysis demonstrated the reliability of N, S-CDs as a probe for rutin detection.

## 3. Conclusions

In this paper, CNC with small particle size and good dispersion was prepared by the sulfuric acid hydrolysis of cellulose from dissolved softwood pulp. CNC was then selected as the green carbon source, and thiourea was employed as the nitrogen-sulfur passivation agent to prepare N, S-CDs with a 27.4% QY value by the hydrothermal method. The prepared N, S-CDs were approximately spherical, uniform in size, with good dispersion. The interaction between oxygen-containing functional groups on the surface of the N, S-CDs and rutin caused the fluorescence quenching of the N, S-CDs solution and provided theoretical support for the detection of rutin. The relationship between log (F_0_/F_1_) value and rutin, with concentrations in the range of 0–40 mg·L^−1^, was well adapted to a linear equation with an LOD value of 0.02 μM. On the basis of the above results, a fluorescence “turn off” sensing system for rutin with N, S-CDs was established, which was simple and rapid for the determination of rutin content. This work highlights the high-value utilization of renewable cellulose to produce CDs, which also provides a new avenue for rapidly detecting rutin.

## 4. Materials and Methods

### 4.1. Materials

Cellulose from dissolved softwood pulp was purchased from Shandong Bohai Industry Co., Ltd. (Boxing, China). Sulfuric acid (H_2_SO_4_, 98% purity, Kermel Analytical Reagent Co., Ltd., Tianjin, China), rutin (95% purity, Aladdin Chemistry Co., Ltd., Shanghai, China), and thiourea (Sinopharm Chemical Reagent Co., Ltd., Shanghai, China) were used without further purification.

### 4.2. Preparation of Cellulose Nanocrystal (CNC)

A total of 3 g cellulose powder was completely soaked into 60 mL H_2_SO_4_ solution (60 wt.%) at 50 °C for 1.5 h under magnetic stirring. After terminating the reaction with an excess of ultrapure water (ten times the volume of the reaction mixture), the white precipitates were derived by centrifugation at 8000 rpm for 10 min and repeatedly washed with ultrapure water 3 times. Finally, the washed suspensions were collected and thoroughly dialyzed against ultrapure water in a dialysis bag (MWCO 3000 Da) until the pH value of the dialysis solution was neutral. The purified CNC was obtained by freeze drying for 72 h.

### 4.3. Preparation of N, S-CDs

Typically, 2 g of CNC were dispersed into 50 mL of ultrapure water and agitated ultrasonically for 30 min to obtain the well-dispersed CNC aqueous solution. Subsequently, 1.6 g of thiourea were slowly added to the CNC solution, which was then transferred into a 100 mL Teflon-lined stainless steel autoclave and heated at 240 °C for 24 h. After being cooled to room temperature, the brown suspension was ultrasonically treated for 30 min with a power of 75 W, followed by centrifugation at 10,000 rpm for 15 min and then it was filtered with a 0.22 μm micropore membrane. Finally, the collected light yellow solution was dialyzed against ultrapure water through a dialysis membrane (MWCO 500 Da) for 24 h. The powder product was obtained by freeze drying the dialysis solution for 72 h, which was then called N, S-CDs.

The comparison CDs (C-CDs) without element passivation were prepared by the hydrothermal treatment of CNC in ultrapure water in the absence of any passivator, and the other preparation and purification processes were same to those for preparing N, S-CDs.

### 4.4. Physicochemical Properties of CNC and N, S-CDs

Scanning electron microscope (SEM) images of the cellulose and CNC samples were obtained by a microscope (JEM-7800F, JEOL, Tokyo, Japan) with an accelerating voltage of 12.5 V. The morphologies of the CNC and N, S-CDs samples were characterized by transmission electron microscopy (JEM-2100, JEOL, Tokyo, Japan) with an accelerating voltage of 200 kV. Fourier transform infrared (FT-IR) spectra of all samples were measured with a Fourier infrared spectrometer (Frontier I, PerkinElmer, Waltham, MA, USA). The XRD patterns of the samples were analyzed with a 6100 X-ray diffractometer (Shimadzu, Kyoto, Japan). The Raman spectrum of N, S-CDs was collected on a LabRAM ARAMIS Raman spectroscope (HORIBA Scientific Jobin, Longjumeau, France). Characterization of X-ray photoelectron spectra (XPS) of N, S-CDs was performed on an ESCALAB250Xi X-ray photoelectron spectrometer (Thermo Fisher Scientific, Waltham, MA, USA).

### 4.5. Photoluminescence Properties of N, S-CDs

Ultraviolet-visible absorbance (UV-vis) spectra were measured with a UV1006M031 UV-vis spectrophotometer (Agilent Technologies Inc., Shanghai, China). The fluorescence quantum yields of N, S-CDs were calculated according to the methods described in previous reports [56]. Each process was repeated at least 5 times. The average value of the results was taken as the final result of the sample.

### 4.6. Fluorescent Detection of Rutin by N, S-CDs

Generally, 1 mL of N, S-CDs solution (0.05 mg·mL^−1^, pH = 5), 200 μL of PBS solution (200 mM, pH = 5), and 1 mL of rutin solution with serial concentrations from 0 mg·mL^−1^ to 125 mg·mL^−1^ were thoroughly mixed, and then ultrapure water was added to the solution to keep the volume at 4 mL. The solutions were left standing for 5 min to achieve equilibrium, and their fluorescence spectra were recorded on a fluorescence spectrophotometer. The lower limit of detection for rutin was calculated, as specified in Equation (1). All of the fluorescence intensities were recorded under an excitation wavelength of 325 nm and an emission wavelength of 405 nm. Each sample was repeated at least 5 times. The average value of the results was taken as the final result of the sample.
(1)LOD=3σ/K
where, K is the slope for the range of the linearity, and σ is the standard deviation of the blank (n > 11).

### 4.7. Detection of Rutin in Real Samples

The lake water samples were collected from a lake at Dalian Polytechnic University, and filtered through a 0.22 μm filter. The human urine samples were obtained from healthy volunteers, and the serum samples were purchased from Aladdin Chemistry Co., Ltd. (Shanghai, China), and then diluted with distilled water (10-fold dilution). Subsequently, 1mL the above samples containing different concentrations of rutin were mixed with 1 mL N, S-CDs solution (0.05 mg·mL^−1^, pH = 5) and 200 μL PBS solution (200 mM, pH = 5), and then ultrapure water was added to the solution to keep the volume at 4 mL. The solutions were left standing for 5 min to achieve equilibrium, and their fluorescence spectra were recorded on a fluorescence spectrophotometer. Each process was repeated at least 5 times. The average value of the results was taken as the final result of the sample.

## Figures and Tables

**Figure 1 molecules-27-08021-f001:**
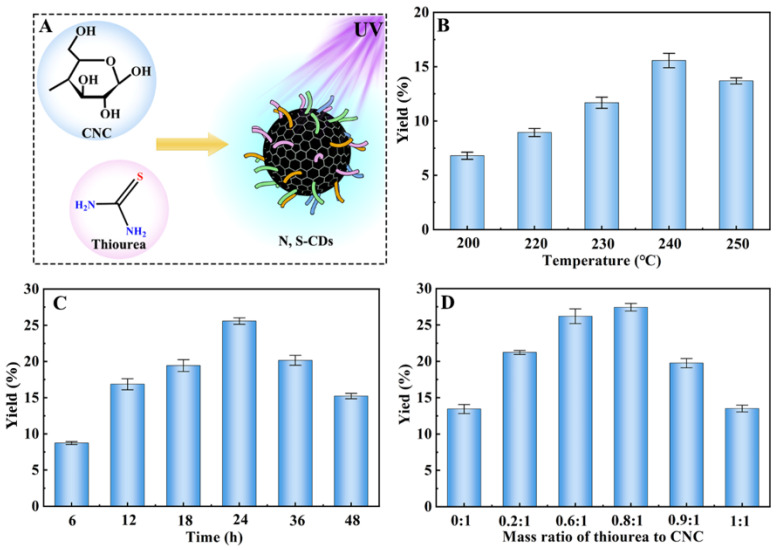
The illustration of the hydrothermal process for preparing N, S-CDs (**A**), the effect of reaction temperatures (**B**), reaction times (**C**), and the mass ratio of thiourea to CNC (**D**) on the QYs of N, S-CDs.

**Figure 2 molecules-27-08021-f002:**
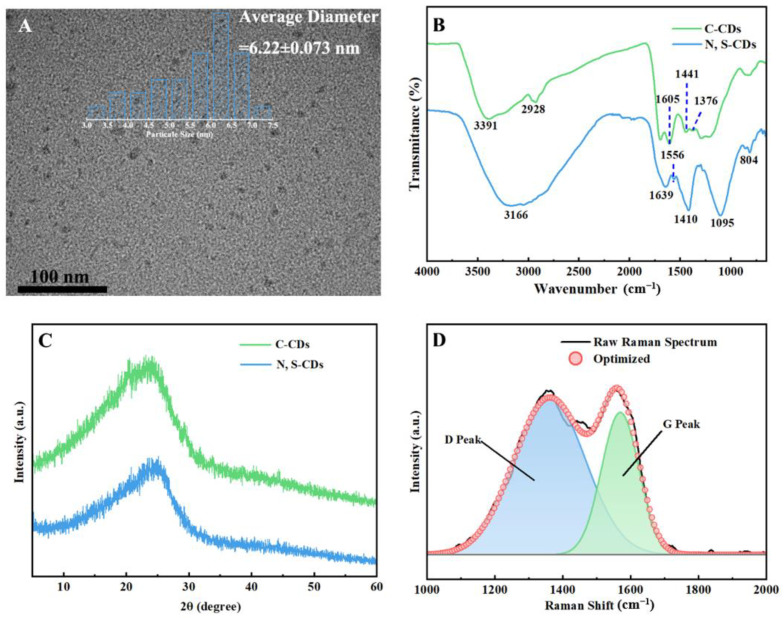
TEM image and histogram of the size distributions of N, S-CDs (**A**), FT−IR spectra of C-CDs and N, S-CDs (**B**), XRD patterns of C-CDs and N, S-CDs (**C**), and Raman spectra of N, S-CDs (**D**).

**Figure 3 molecules-27-08021-f003:**
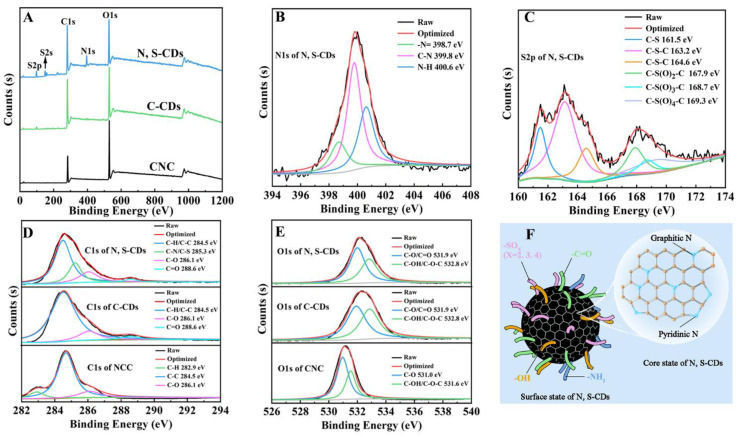
XPS spectra of CNC, C-CDs and N, S-CDs (**A**); high-resolution N1s (**B**) and S2p (**C**) spectra of N, S-CDs; high-resolution C1s (**D**) and O1s (**E**) spectra of CNC, C-CDs and N, S-CDs; the schematic diagram (**F**) of the surface and core state of N, S-CDs.

**Figure 4 molecules-27-08021-f004:**
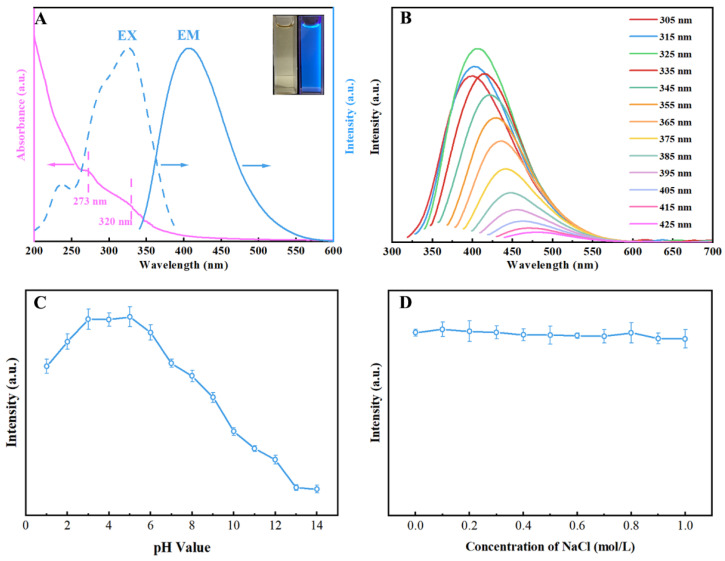
UV−vis absorption spectra of N, S-CDs and the excitation/emission fluorescence spectra of N, S-CDs. The top right inset shows the digital photographs of the N, S-CDs solutions under daylight and UV radiation at 365 nm (**A**); fluorescence emission spectra of N, S-CDs at different ranges of excitation wavelengths (**B**); fluorescence intensity of N, S-CDs at different pH values (**C**); and various concentrations of NaCl ionic solutions (**D**).

**Figure 5 molecules-27-08021-f005:**
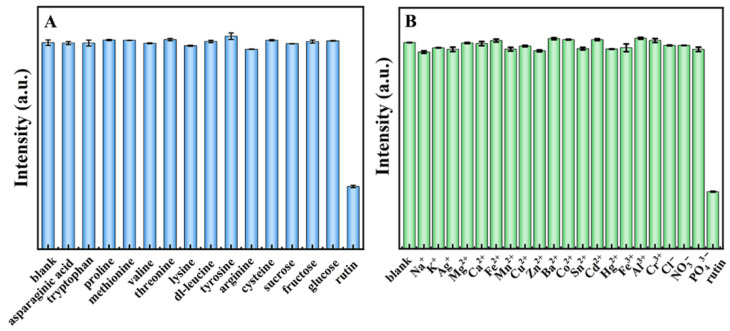
Effects of coexisting different amino acids (100 μM) (**A**) and metal ions (100 μM) (**B**) with rutin (45 mg·L^−1^) on the fluorescence intensity of N, S-CDs.

**Figure 6 molecules-27-08021-f006:**
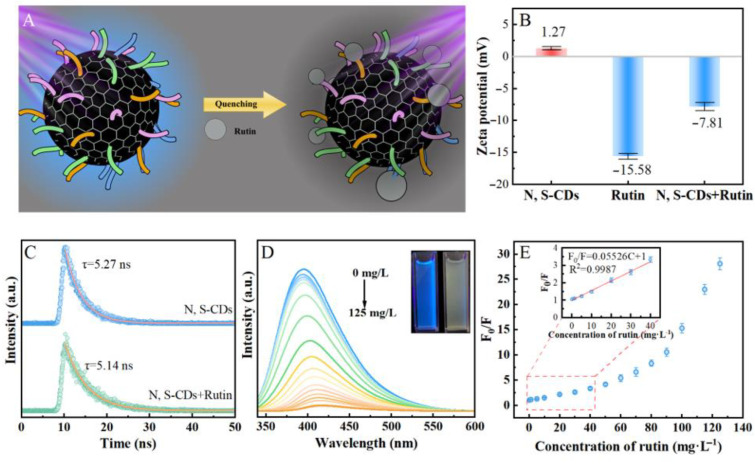
The schematic diagram for the fluorescence quenching process of N, S-CDs in the presence of rutin (**A**); the zeta potential of fluorescence quenching system (**B**); fluorescence lifetime of N, S-CDs in the absence and presence of rutin (**C**); fluorescent emission spectra of the N, S-CDs in the presence of different amounts of rutin (**D**); the dependence of the F_0_/F value on the different concentration of rutin (the inset was the Stern−Volmer plots) (**E**).

**Table 1 molecules-27-08021-t001:** The contents of C, O, N, and S elements in N, S-CDs according to XPS analysis.

Elements	Samples		
	CNC	C-CDs	N, S-CDs
C	54.89%	66.28%	58.24%
O	44.05%	28.99%	26.90%
N	-	-	7.90%
S	-	-	6.96%

**Table 2 molecules-27-08021-t002:** Comparison of the other reported optical materials for rutin detection.

Materials	LODs(μM)	K_SV_ (M^−1^)	Ref.
copper nanoclusters	0.02	71,500	[30]
copper nanoclusters	0.01	38,000	[51]
ZnS quantum dots	0.06	-	[52]
BA-CdTe@MIPs QDs	0.06	95,000	[53]
silicon nanoparticles	0.01	4940	[54]
silicon nanoparticles	0.04	-	[55]
N, S-CDs	0.02	55,260	This work

**Table 3 molecules-27-08021-t003:** Results of the rutin recoveries in lake water, human urine, and serum samples.

Samples	Rutin Spiked (mg·L^−1^)	Measured (mg·L^−1^, n = 5)	Recovery (%)	RSD (%)
Lake water 1	1	0.97	96.5	2.8
Lake water 2	15	15.24	101.6	2.4
Lake water 3	40	38.85	97.1	3.2
Human urine 1	1	1.04	104.3	1.9
Human urine 2	15	14.68	97.9	2.1
Human urine 3	40	40.93	1.02	1.6
Serum 1	1	0.95	96.3	3.1
Serum 2	15	15.83	104.2	3.4
Serum 3	40	39.17	97.9	2.3

## Data Availability

Not applicable.

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
