# Peer review of "Preparation of Nitrogen and Sulfur Co-Doped Fluorescent Carbon Dots from Cellulose Nanocrystals as a Sensor for the Detection of Rutin"

_molecules, 2022, doi:10.3390/molecules27228021_

Round 1
Reviewer 1 Report
This work reports on the development of nitrogen, sulfur co-doped fluorescent carbon dots (N, S-CDs) using cellulose nanocrystals (CNC) for the detection of rutin. My comments are as follows:
1. Suggest highlighting the novelty/contribution of the work in the abstract section.
2. In the methodology section, suggest dividing the characterization part based on the properties of characterization for easy understanding. For example, "molecular characterization", "surface characterization" and so forth.
3. Suggest including the selectivity test for the sensor performance.
4. I notice there are a lot of inconsistent fonts used throughout the document. Kindly standardize the font accordingly.
Reviewer 2 Report
There are several serious omissions. The novelty is not high.
Below, you can find my specific comments:
1. The introduction should have a focus on current analytical challenges for rutin determination.
2. Authors should explain why they have decided to use cellulose nanocrystals as precursors and not any other form of cellulose.
3. Give error bars in Figures 1BCD, 4CD and 6C.
4. Show a Stern-Volmer plot of the type
(Iâ‚’/If) = 1 + Ksv[quencher]
(a) in order to see whether quenching is linear, or upward curved, or downward curved,
(b) in order to see whether quenching is static or dynamic, and
(c) in order to express quenching efficiency by a number.
5. The putative mechanism is not justified by any results. The authors should record the quenching at different temperatures and study the fluorescence lifetime.
6. Some experimental data lack (relative) standard deviations. Give averaged data for important experimental data along with standard deviations (±) and the number of experiments (n = ?).
7. The authors do not adequately explain why a reader of the article should use the new method rather than an existing method that possibly works quite well and is simpler. They should provide a Table x, with an overview on recently reported nanomaterial-based optical methods for the determination of rutin; with columns on **Materials used; **Figures of merit (such as LODs and specificity); and **a column giving the respective references.
8. Real sample analysis: The authors provide data on the analysis of a samples including RSDs and recovery studies.
9. Replace "correlation coefficient R2 = 0.9975" with "coefficient of determination R2 = 0.9975"
10. The lowest concentration in the linear range cannot be 0. The authors should provide a rational first point of the linear range based on the limit of quantification.
11. Experimental. Give a protocol on how to perform the assay with a real sample, step by step. How much sample shall be taken? How shall it be treated? Which buffer and reagents (how many mg or mL and in which concentration) have to be added? How are analytical data generated?
Reviewer 3 Report
In the present manuscript entitled “Preparation of nitrogen, sulfur co-doped fluorescent carbon dots from cellulose nanocrystals as a sensor for detection of rutin”, the authors proposed a new method for carbon dots preparation from cellulose nanocrystals and as obtained carbon dots were applied as a sensor for rutin. Carbon dots synthesis part is novel and well explored by the authors. However, the rutin detection part is poorly studied. The authors failed explain the possible mechanism of detection, and there is no real sample analysis. Hence I suggest a major revision.
Major comments
1. What is the importance of rutin detection ? Authors should describe it in few lines in introduction section or results section.
2. There is no clear data and explanation on rutin detection mechanism. The authors provided absorption spectra for this. However no detectable changes were observed. The authors should study the changes in fluorescence lifetime and zeta potential.
3. Detection performance should be compared with previous reports in a table.
4. Real sample analysis should be performed
5. Authors can refer citation 41 and 44 of their manuscript for detection mechanism studies and real sample analysis.
6. Photostability studies of NSCDs should be added
Minor comments
1. In abstract, line 24, units were expressed as mg.L-1 and mg/L. Authors should use a uniform expression throughout the manuscript
2. “C-CDs or CDs” use a uniform abbreviation throughout the manuscript
3. Appropriate citation should be provided for CNC preparation method
4. Low magnification TEM image of CNC should be provided
5. In fig 3 caption, there is a wrong assignment for C 1s spectra.
Round 2
Reviewer 1 Report
The authors have made amendments accordingly and the manuscript is improved tremendously. I recommend it be accepted by molecules.
Reviewer 2 Report
The authors have taken into consideration the points raised by the reviewing. Although the originality is average, overall the quality of manuscript is such that it can be accepted.
Reviewer 3 Report
The authors well addressed all my comments.